physical chemistry/green chemistry

manganese oxide, electrocatalyst, water splitting

**Author for correspondence:**
Ryuhei Nakamura
e-mail: ryuhei.nakamura@riken.jp

†Present address: Department of Materials Science and Engineering, School of Materials and Chemical Technology, Tokyo Institute of Technology, S7-9, 2-12-1 Ookayama, Meguro-ku, Tokyo 152-8552, Japan.

This article has been edited by the Royal Society of Chemistry, including the commissioning, peer review process and editorial aspects up to the point of acceptance.

# Electrochemical characterization of manganese oxides as a water oxidation catalyst in proton exchange membrane electrolysers

Toru Hayashi[1,2], Nadège Bonnet-Mercier[2], Akira Yamaguchi[2,†], Kazumasa Suetsugu[3] and Ryuhei Nakamura[2,4]

[1]Department of Applied Chemistry, The University of Tokyo, 7-3-1 Hongo, Bunkyo-ku, Tokyo 113-8656, Japan
[2]Biofunctional Catalyst Research Team, RIKEN Center for Sustainable Resource Science (CSRS), 2-1 Hirosawa, Wako, Saitama 351-0198, Japan
[3]Tosoh Corporation, 3-8-2 Shiba, Minato-ku, Tokyo 105-8623, Japan
[4]Earth-Life Science Institute (ELSI), Tokyo Institute of Technology, 2-12-1 Ookayama, Meguro-ku, Tokyo 152-0033, Japan

AY, 0000-0002-3550-4239; RN, 0000-0003-0743-8534

The performance of four polymorphs of manganese (Mn) dioxides as the catalyst for the oxygen evolution reaction (OER) in proton exchange membrane (PEM) electrolysers was examined. The comparison of the activity between Mn oxides/carbon (Mn/C), iridium oxide/carbon (Ir/C) and platinum/carbon (Pt/C) under the same condition in PEM electrolysers showed that the $\gamma$-MnO$_2$/C exhibited a voltage efficiency for water electrolysis comparable to the case with Pt/C, while lower than the case with the benchmark Ir/C OER catalyst. The rapid decrease in the voltage efficiency was observed for a PEM electrolyser with the Mn/C, as indicated by the voltage shift from 1.7 to 1.9 V under the galvanostatic condition. The rapid deactivation was also observed when Pt/C was used, indicating that the instability of PEM electrolysis with Mn/C is probably due to the oxidative decomposition of carbon supports. The OER activity of the four types of Mn oxides was also evaluated at acidic pH in a three-electrode system. It was found that the OER activity trends of the Mn oxides evaluated in an acidic aqueous electrolyte were distinct from those in PEM electrolysers, demonstrating the importance of the evaluation of OER catalysts in a real device condition for future development of noble-metal-free PEM electrolysers.

# 1. Introduction

The development of storage methods of renewable energy has been an urgent issue for addressing the intermittency problem impeding the increase in renewable energy usage. One of the promising methods is the production of chemical fuels by the electrochemical reduction of protons to hydrogen gas or carbon dioxide to hydrocarbon [1–4]. In the development of electrolysers, however, the sluggish oxygen evolution reaction (OER) via water oxidation ($2H_2O \rightarrow O_2 + 4H^+ + 4e^-$), where four-proton and four-electron transfer has to be managed, often becomes a bottleneck for the efficiency and stability of the systems [5]. The most efficient catalysts for the OER currently known are iridium (Ir)-based or ruthenium (Ru)-based catalysts [6–10], whose large-scale use is fundamentally hampered by low production of these metals. According to the International Energy Agency, global energy demand will increase from 18 TW in 2014 to 24 or 26 TW in 2040 in 'new policies' or 'current policies' scenario, respectively [11]. Under such circumstances, the limited and inelastic supply of Ir and Ru prohibit the terawatt-level deployment of electrolysers, as reported by Vesborg & Jaramillo [12]. While approximately 300 tons $TW^{-1}$ of OER catalysts are required for electrolysers, the annual production of Ir, for example, is less than 9 tons [12]. The annual production of Ru is also in the same order [12]. Thus, the development of efficient OER catalysts composed of earth-abundant elements has been regarded as a fundamental challenge to be solved for the production of chemical fuels by renewable energy [10,13–18].

Up until now, three types of electrolysers have been mainly developed for electrochemical water splitting: alkaline electrolysers [19], proton exchange membrane (or polymer electrolyte membrane; PEM) electrolysers [20] and solid oxide electrolysers [21,22]. Among them, PEM electrolysers possess great advantages for the energy conversion systems from renewable to chemical energy, due to the high voltage efficiency, rapid response to input changes, high current density, usage of non-corrosive liquid and relatively low working temperature (less than 100°C) [20]. These advantages are directly or indirectly related to current standard PEMs with low gas permeation rate and high proton conductivity, typically composed of perfluorinated sulfonic acid polymers [20,23]. However, because of the corrosive acidic environment for catalysts provided by PEMs, noble metal OER catalysts have been used for PEM electrolysers, while the reasons why noble metals can function as an active and stable catalyst in acidic environments remain largely unknown [9,10].

One strategy to address the problem of the corrosive environment is to use alkaline exchange membranes (AEMs) [24–26], and a number of efficient non-noble metal OER catalysts working in strongly alkaline conditions have been developed [27]. There has recently been significant progress in the development of AEMs [24], yet the ionic conductivity and stability of AEMs still need to be largely improved to rival those of PEMs such as Nafion. Also, the lower mobility of $OH^-$ than that of $H^+$ is an intrinsic problem for AEMs [23,28].

For the terawatt-level deployment of PEM electrolysers, the catalysts are required to be composed of earth-abundant elements. Most studies for the development of the OER catalysts functional in acidic conditions have been focused on materials containing Ir or Ru [6–8] and how to decrease their noble metal content [6,7,29–35]. The OER activity per amount of noble metal has already been successfully enhanced, for example, by using mixed oxides with less expensive metals such as nickel [29,30], tungsten [31] or tin [32], or nano-structuring [6,33–35]. Only very recently, earth-abundant OER catalysts which can operate at acidic pH are beginning to be reported. Among them, especially, manganese (Mn)-based materials are considered as promising substitutes for Ir-based or Ru-based catalysts, as, even in acidic conditions, there is a possibility that Mn oxide phases are preserved without dissolution to $Mn^{2+}$ at anodic potential, keeping their function as OER catalysts [36]. Nocera and co-workers [36] reported that, in acidic conditions (pH = 0.1), a Mn oxide can functionally stably catalyse the OER by virtue of the oxidative electrodeposition of dissolved Mn ions. Stephens and co-workers [37] reported that titanium atoms on a Mn oxide can mitigate the dissolution in acidic conditions (pH 1) without large decrease in the OER activity. Fluorine-doped copper Mn oxides were developed by Kumta and co-workers [38] which showed excellent activity and stability for the OER in 0.5 M $H_2SO_4$ aqueous solution. In addition to these Mn-oxide-based catalysts, cobalt iron cyanide [39], Janus Co/CoP nanoparticles [40], $MoS_2$, $TaS_2$ [41], $Co_3O_4$ [42] and barium salt of a cobalt-phosphotungstate polyanion [43] were recently introduced as the catalysts for the OER in acidic conditions. They maintained their activity for hours in acid. Nocera and co-workers [44] also reported that the stability of a cobalt oxide in acid prepared by electrodeposition can be enhanced by co-deposition with other metal ions to obtain $CoMnO_x$ or $CoFePbO_x$ without changes in the OER activity. Lewis and co-workers [45] reported a nickel Mn antimonate as a stable OER catalyst in aqueous 1.0 M $H_2SO_4$. Although the works introduced above represent significant progress in the

development of acid-stable earth-abundant OER catalysts, further study will be required for the development of the PEM electrolysers suitable for the large-scale energy conversion and storage systems. Especially, the research on PEM electrolysers has been limited to those with noble metal catalysts, and no earth-abundant OER catalysts have been evaluated in PEM electrolysers.

Herein, Mn oxide samples were evaluated as OER catalysts in PEM electrolysers. The emphasis has been placed on the comparison of the activity between Mn oxide and Ir, as well as platinum (Pt) catalysts in the same condition in PEM electrolysers. PEM electrolysers were constructed with the Mn oxide catalysts synthesized by the methods for industrial processes to produce electrolytic Mn dioxides (EMD) and chemical Mn dioxides (CMD), and the water splitting efficiency and stability were compared with those of benchmark Ir/C catalysts. Pt/C was also used as a reference. Although carbon supports can have a problem in stability due to electrochemical oxidation [46–48], they have been used for short-term tests or the characterization of electrocatalysts [20]. Also, the use of an Ir/C catalyst (Premetek), which is widely used as a benchmark OER catalyst in the field of water electrolysis in acidic and neutral conditions, allows us to compare the performance of EMD and CMD to a wealth of earth-abundant OER catalysts. To further evaluate the characteristics of the anodes, the OER activity of the catalysts in the presence of Nafion ionomer was measured in a three-electrode system by means of linear sweep voltammetry (LSV) and Tafel plots. The activity of the catalysts was also evaluated in acidic aqueous electrolyte. Based on the current density–voltage curves of the PEM electrolysers and the results of electrolysis in a galvanostatic condition, possible degradation mechanisms will be discussed to address the stability problems of the PEM electrolysers.

# 2. Material and methods

## 2.1. Synthesis of Mn oxides

$\alpha$-MnO$_2$: Electrodeposition of a Mn oxide on a titanium anode was conducted with the aqueous electrolyte containing 0.502 M of MnSO$_4$ and 3.0 M of (NH$_4$)$_2$SO$_4$ at 8 mA cm$^{-2}$ and 96°C for 25 h. The concentration of NH$_4^+$ and SO$_4^{2-}$ was maintained to be 3.0 M and 0.31 M, respectively, by continuous supply of the aqueous solution of MnSO$_4$ and (NH$_4$)$_2$SO$_4$. The deposit was ground, washed with water and dried to obtain the final sample.

$\beta$-MnO$_2$: Electrodeposition of a Mn oxide on a titanium anode was conducted in a bath with an H$_2$SO$_4$–MnSO$_4$ mixture at 7 mA cm$^{-2}$ and 96°C for 10 days, while maintaining the concentration of SO$_4^{2-}$ to be 0.33 M by continuous addition of 0.85 M MnSO$_4$ aqueous solution. The resultant deposit was ground for the average diameter of secondary particles to be 40 µm, washed with water, neutralized and dried by flash drying. The sample was obtained by calcination of the dried sample at 420°C for 36 h.

$\delta$-MnO$_2$: 2.5 l of the aqueous solution with 0.100 M of KMnO$_4$ and 2.50 M of H$_2$SO$_4$ was added to 1 l of the aqueous solution containing 0.552 M of Mn ion and 2.55 M of SO$_4^{2-}$ (an acidic MnSO$_4$ solution), followed by stirring at 30°C for 24 h. After separated by filtration, the black deposit was washed by dispersing it in 500 ml of pure water for 1 h twice. The washed deposit was dispersed again in 500 ml of pure water and neutralized by the addition of 1 M NaOH aqueous solution until the pH of the slurry became 5.6. The final sample was obtained by filtration and drying.

$\gamma$-MnO$_2$: During the flash drying in the synthesis of $\beta$-MnO$_2$, the fine powder generated by overgrinding was collected by a bag filter of a dust collector and used as the sample.

## 2.2. Characterization of physical properties of samples

Crystal structures were identified by an X-ray diffraction (XRD) diffractometer (SmartLab, Rigaku) with Cu K$\alpha$ radiation ($\lambda = 1.54059$ Å) using a voltage and current of 45 kV and 200 mA, respectively.

The average valency of Mn was measured by titration and the evaluation of purity. First, 0.200 g of the Mn oxide sample was dissolved in the mixture of 10 ml of 0.3 M (COOH)$_2$ aqueous solution and 20 ml of 9 M H$_2$SO$_4$ aqueous solution at 70°C. The solution was titrated by 0.6328 mM KMnO$_4$ aqueous solution. The purity of Mn$^{4+}$ was calculated by the following equation:

$$\text{Purity of Mn}^{4+}(\%) = (B - A) \times 2.1735, \tag{2.1}$$

where $A$ and $B$ are the volume of 0.6328 mM KMnO$_4$ solution in millilitre required for blank test and the titration, respectively. The purity of total Mn was measured by inductively coupled plasma

**Figure 1.** Schematic of the preparation of MEA and the structure of PEM electrolysers used in this study.

atomic emission spectroscopy (ICP-AES). Finally, the average Mn valency was calculated by the following equation:

$$\text{Average valency of Mn } = \frac{\text{Purity of Mn}^{4+}(\%) \times 63.19}{\text{Purity of total Mn } (\%) \times 2}. \tag{2.2}$$

The diameter of primary and secondary particles was evaluated by a scanning electron microscope (SEM) (S-4800, Hitachi) and a laser-based particle size analyser (Microtrac HRA, Honeywell), respectively. For the measurement of the diameter of secondary particles, 0.5 g of the sample was used after dispersion in 50 ml of pure water by sonication for 10 s. The 1.33 and 2.20 were used as the refractive index of pure water and Mn oxide, respectively.

Brunauer–Emmett–Teller (BET) surface area was measured by the flowing gas method using an automatic analyser (FlowSorb III, Shimadzu), using nitrogen gas and the single-point method. The sample was heated at 150°C for 40 min for degassing before the measurement.

## 2.3. PEM electrolysis

The anodes for PEM electrolysers were prepared as follows. The Mn oxide sample was first blended with carbon black (Vulcan XC-72) with a weight ratio of one to nine in an agar mortar. The mixture was then transferred in a container with 50 ml of ethanol and zirconia balls (0.3 mm in diameter) and subjected to ball milling for 24 h at 40 r.p.m. After that, the zirconia balls were removed using a sieve to obtain a slurry containing the Mn oxide, carbon black and ethanol. The weight ratio of the Mn oxide sample to the total weight of the Mn oxide and the carbon black was 5.9%, 11.2%, 10.3% and 11.6% for α-MnO$_2$, β-MnO$_2$, δ-MnO$_2$ and γ-MnO$_2$, respectively. By using the slurry, Ir/C (20 wt% Ir on Vulcan XC-72, Premetek) or Pt/C (20 wt% Pt on Vulcan XC-72, Fuel Cell Earth), 1.5 ml of catalyst inks containing 15 mg of the total amount of the catalyst (Mn oxide, Ir or Pt) and the carbon black and 50 µl of 10 wt% Nafion aqueous dispersion (no. 527106, Sigma-Aldrich) were then prepared (figure 1). Finally, 500 µl of the catalyst ink was coated on a carbon paper (TGP-H-060, Toray) and dried in air.

The cathode was prepared in the same way as the anode using Pt/C as the catalyst for hydrogen evolution. Nafion 117 membrane was washed and protonated by boiling sequentially in 3% H$_2$O$_2$ aqueous solution, pure water, 1 M H$_2$SO$_4$ aqueous solution and pure water, for 1 h each.

The pretreated Nafion 117 membrane was then sandwiched between the anode and the cathode with the catalyst-coated side in contact with the Nafion membrane. The stacking was then closely connected by a hot press at 135°C and a mould clamping force of 600 kg for 10 min. The membrane electrode assembly (MEA) thus obtained was incorporated into the chassis of a PEM electrolyser (3036, FC-R&D), which does not have heating devices, and used as a two-electrode system for subsequent measurement (figure 1). The measurement was conducted using a potentiostat (HZ-7000, Hokuto Denko) at room temperature. The cell resistance was also measured by the potentiostat using a square wave of direct current (superposition of direct current of two potentials).

## 2.4. Preparation of electrodes (other than those for PEM electrolysers)

The working electrodes with or without Nafion ionomer for the measurement in three-electrode systems were fabricated as follows.

The electrodes for the evaluation of the catalysts in the presence of Nafion ionomer were prepared, by reference to a typical method for the evaluation of catalysts for PEM-based devices [49]. Slurries containing the $\gamma$-$MnO_2$, carbon black and ethanol were prepared in the same way as described above. The weight ratio of the $\gamma$-$MnO_2$ and carbon black in the resultant slurry was 9.9 : 90.1. By using the slurry, the Ir/C or the Pt/C, 1.5 ml of catalyst inks containing 3 mg of the catalyst (the $\gamma$-$MnO_2$, Ir or Pt) and 50 µl of the 10 wt% Nafion aqueous dispersion were then prepared. Finally, 7 µl of the ink was dropped on a glassy carbon rotating disc electrode (RDE) (diameter: 5 mm) (HR2-D1-GC5, Hokuto Denko) and dried in air.

Particulate Mn oxide film electrodes without Nafion were prepared by a spray-coating method onto fluorine-doped tin oxide (FTO) substrates, as previously described [16]. The amount of deposited Mn oxides was approximately 0.14 mg cm$^{-2}$. About 75 mg of the sample was ground in an agate mortar for 5 min and suspended in 100 ml of highly pure Milli-Q water (18 MΩ cm$^{-1}$) by a sonicator (Q700, QSonica). The suspension was sprayed by a spray gun (ST-6, Fuso Seiki Co., Ltd) onto clean conducting FTO-coated glass substrates (SPD Laboratory, Inc.) heated at 200°C. The electrodes were gently washed with the highly pure Milli-Q water and calcined in air for 4 h at 500°C.

## 2.5. Electrochemical measurement

Electrochemical measurement using three-electrode systems was also conducted with the potentiostat at room temperature. A Pt wire and Ag/AgCl/sat. KCl were used as the counter and reference electrode, respectively. Considering that pure neutral water is usually electrolysed in PEM electrolysers, neutral electrolyte, 0.5 M $Na_2SO_4$ aqueous solution (pH 7.5 adjusted by NaOH and $H_2SO_4$), was used for the evaluation of Nafion-containing working electrodes. The RDE was rotated at 1600 r.p.m. Ohmic resistance was corrected based on the resistance before the measurement. Tafel plots were made by plotting steady-state current at each potential. The highest potential was kept for 1500 s to avoid the effect of pseudo-capacitance. At other potentials, the steady-state current was measured after 240 s.

# 3. Results and discussion

## 3.1. Material characterization

Four types of Mn oxides were synthesized by industrial processes to produce EMD and CMD, of which main applications currently include dry cells and the raw material for the anode of lithium-ion batteries and ferrites. The crystal structures of the samples were analysed by XRD, and identified as $\alpha$-$MnO_2$, $\beta$-$MnO_2$, $\delta$-$MnO_2$ and $\gamma$-$MnO_2$ (figures 2 and 3). The microscopic order of the structure and electrochemical functions of $\gamma$-$MnO_2$ is known to be affected by synthetic conditions. Thus, the structure of the $\gamma$-$MnO_2$ was analysed in detail. The XRD pattern of the $\gamma$-$MnO_2$ sample showed the existence of extensive microtwinning in the ramsdellite 021/061 planes, which is the typical structure of EMD [50] prepared at high current densities [51]. The peaks at $2\theta = 22.0°$, 37.0°, 42.4°, 56.2° and 67.2° were assigned to (110) plane of orthorhombic ramsdellite Pbnm structure and (100), (101), (102) and (110) planes of the pseudo-hexagonal cell, respectively [51]. The structure of $\gamma$-$MnO_2$ is a random intergrowth of pyrolusite (De Wolff disorder) (with $1 \times 1$ tunnels) in the ramsdellite matrix (with $1 \times 2$ tunnels) (figure 3). The ratio of pyrolusite domains calculated by Chabre and Pannetier's method was 0.35 [51].

SEM images of the Mn oxide samples revealed the morphology and distribution of the particles, from which the Krumbein diameter was calculated (figure 4 and table 1). Nanowire morphology of $\delta$-$MnO_2$ was evident (figure 4c), and the aspect ratio of the samples $\alpha$-$MnO_2$, $\beta$-$MnO_2$ and $\gamma$-$MnO_2$ was smaller (figure 4a,b,d). Including BET surface area and bulk density, the result of the analysis of physical properties of the samples was summarized in table 1.

## 3.2. PEM electrolysis

The samples were evaluated as OER catalysts in PEM electrolysers and their activity was compared with benchmark Ir/C, or Pt/C (20 wt% metal on carbon black Vulcan XC-72) catalysts using the same

**Figure 2.** XRD patterns of the four Mn oxides synthesized by industrial processes and evaluated herein: (*a*) α-MnO$_2$, (*b*) β-MnO$_2$, (*c*) δ-MnO$_2$ and (*d*) γ-MnO$_2$. Standard XRD patterns from JCPDS cards are also presented as references.

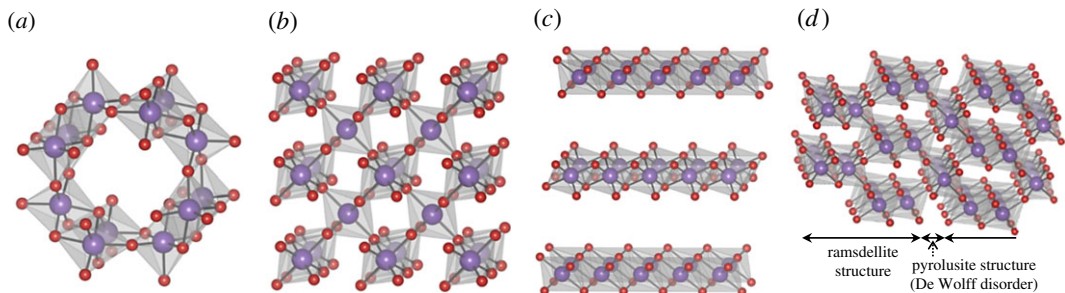

**Figure 3.** Schematic of the crystal structures of (*a*) α-MnO$_2$, (*b*) β-MnO$_2$, (*c*) δ-MnO$_2$ and (*d*) γ-MnO$_2$.

conducting support (a carbon paper: TGP-H-060, Toray). Here, to compare the catalysts in PEM electrolysers, the current density was normalized by the total weight of the catalyst and carbon black. Again, although carbon supports can have a problem in stability due to electrochemical oxidation [46–48], they have been used for short-term tests or the characterization of electrocatalysts [20]. The Ir/C from Premetek was adopted as the benchmark Ir OER catalyst, as it has been used as a highly active benchmark catalyst in the literature [52–55]. Pt/C was also adopted in this study, as PEM

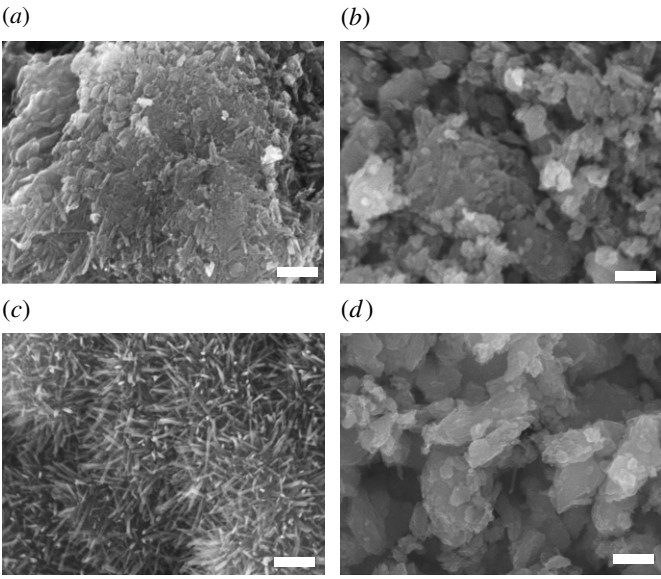

**Figure 4.** SEM images of the four Mn oxides synthesized by industrial processes and evaluated herein: (*a*) α-MnO$_2$, (*b*) β-MnO$_2$, (*c*) δ-MnO$_2$ and (*d*) γ-MnO$_2$ (scale bar, 200 nm).

**Table 1.** Physical properties of the four types of Mn oxides synthesized by industrial processes and evaluated herein.

| | avg. Mn valency | diameter of primary particles (nm) | | | avg. diameter of secondary particles (μm) | Brunauer– Emmett–Teller (BET) surface area (m$^2$ g$^{-1}$) | bulk density (g cm$^{-3}$) |
| | | avg. length of short axes | avg. length of long axes | avg. | | | |
|---|---|---|---|---|---|---|---|
| α-MnO$_2$ | 3.92 | 16 | 80 | 48 | 22 | 95 | 1.4 |
| β-MnO$_2$ | 3.99 | 20 | 60 | 40 | 40 | 13.6 | 2.2 |
| δ-MnO$_2$ | 3.86 | 16 | 150 | 83 | 9.7 | 230 | 0.5 |
| γ-MnO$_2$ | 3.94 | 12 | 34 | 23 | 0.6 | 40 | 1.5 |

electrolysers with a Pt/C OER catalyst connected with solar cells realized solar-to-hydrogen energy conversion efficiency of 24.4% [56,57]. In the present study, the Mn oxides were mixed with carbon black Vulcan XC-72 and Nafion ionomer before loading on the carbon paper anode. The total amount of the catalyst and carbon black loaded on the anodes was fixed to 5 mg (figure 1).

Figure 5 shows the current density–voltage curves with the mixture of an Mn oxide and carbon black in a PEM electrolyser at room temperature. With the Mn oxides examined herein, the onset voltage defined at 0.5 mA cm$^{-2}$ was 1.65–1.66 V (figure 5 and table 2). At 2 V, the current density reached 3.17–3.98 mA cm$^{-2}$ (figure 5 and table 2). Among the Mn oxide samples, the γ-MnO$_2$ showed the highest current density at 2 V (figure 5 and table 2). Investigation of the reason for the highest activity and the optimization of the material are currently underway. However, it is to be noted that γ-MnO$_2$ has a stable potential window in an acidic condition, where the OER can be catalysed efficiently by suppressing the corrosion reaction [58]. Therefore, we expect that the existence of such a stable potential window in acid may contribute the high OER activity of γ-MnO$_2$. With Ir/C, the onset voltage was 1.51 V, and the current density at 2 V was 6.02 mA cm$^{-2}$, which was 1.5 times larger than the value with the γ-MnO$_2$ (figure 5 and table 2). On the other hand, the onset voltage and the current density at 2 V in the case with Mn oxide samples were comparable to those with Pt/C (1.67 V and 3.60 mA cm$^{-2}$) (figure 5 and table 2).

## 3.3. OER activity in a three-electrode system

For further evaluation of the Mn oxide OER catalyst, LSV and Tafel analysis of the γ-MnO$_2$ sample were conducted in the presence of Nafion ionomer using a three-electrode system with Ag/AgCl/sat. KCl and

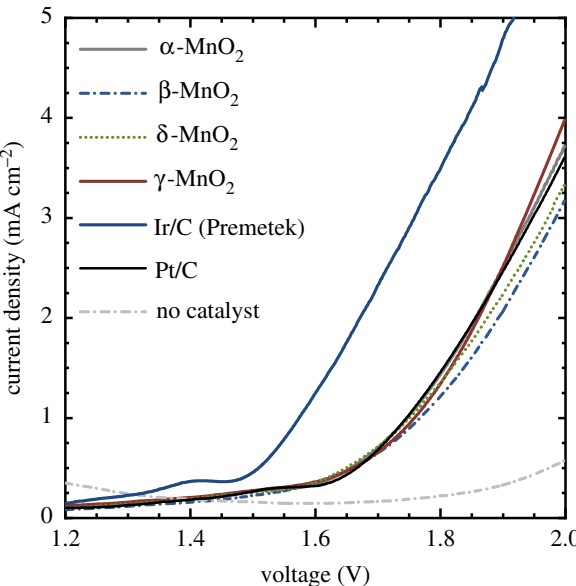

**Figure 5.** Current density–voltage curves of PEM electrolysers with the Mn oxide samples synthesized by industrial processes and evaluated herein, Ir/C (Premetek), or Pt/C for OER catalysts (scan rate: 5 mV s$^{-1}$). The results of the third scans, after the curves became stable, are shown.

**Table 2.** Summary of the voltage at 2 mA cm$^{-2}$ for the PEM electrolysers and the current density for the PEM electrolysers at 2 V and an acidic aqueous electrolyte (pH 0.2) 2 V versus RHE (discussed later: figure 7).

| | voltage (V) | current density (mA cm$^{-2}$) | |
| --- | --- | --- | --- |
| catalyst | PEM electrolyser (at 2 mA cm$^{-2}$) | PEM electrolysers (at 2 V) | aqueous electrolyte (pH 0.2) (at 2 V versus RHE) |
| $\alpha$-MnO$_2$ | 1.856 | 3.72 | 11.5 |
| $\beta$-MnO$_2$ | 1.893 | 3.17 | 4.74 |
| $\delta$-MnO$_2$ | 1.875 | 3.33 | 2.92 |
| $\gamma$-MnO$_2$ | 1.861 | 3.98 | 2.68 |
| Ir/C (Premetek) | 1.671 | 6.02 | — |
| Pt/C | 1.856 | 3.60 | — |

compared to the results with benchmark Ir/C or Pt/C. Here, by reference to a typical procedure to evaluate catalysts in the presence of Nafion ionomer [49], the working electrode was prepared by loading the catalyst ink containing the $\gamma$-MnO$_2$ and carbon black, Ir/C, or Pt/C and Nafion ionomer on a glassy carbon electrode.

As shown in the LSV curves (figure 6a), the OER activity of the $\gamma$-MnO$_2$ and Pt/C was comparable, which is consistent with the comparable water splitting efficiency of the PEM electrolysers with these OER catalysts (figure 5). The slopes of Tafel plots were 220 mV, 104 and 181 mV dec$^{-1}$ for the $\gamma$-MnO$_2$, Ir/C and Pt/C, respectively (figure 6b and table 3). The typical values of Tafel slopes for the OER on MnO$_2$ and IrO$_2$ in neutral aqueous electrolyte are 120 mV dec$^{-1}$ [36,59] and 30–60 mV dec$^{-1}$ [6,9], respectively (table 3). The higher slope values in the presence of Nafion ionomer may reflect slower diffusion of reaction species [60]. The Tafel slope much larger than 120 mV dec$^{-1}$ is an indication that single electron transfer with a very high symmetry factor ($\beta$), or a chemical process occurring from the resting state of the catalyst is the turnover-limiting step [13]. As the one-electron oxidation of Mn$^{2+}$ was the turnover-limiting step for the OER by Mn oxides when Mn$^{3+}$ disproportionates to Mn$^{2+}$ to Mn$^{4+}$ [61], it is expected that the turnover-limiting step in this case is the one-electron oxidation of Mn$^{2+}$ with a very high value of $\beta$. The Tafel slopes previously observed for the OER by Pt catalysts were also larger than 120 mV dec$^{-1}$ (table 3), probably due to the formation of surface oxide layers [6].

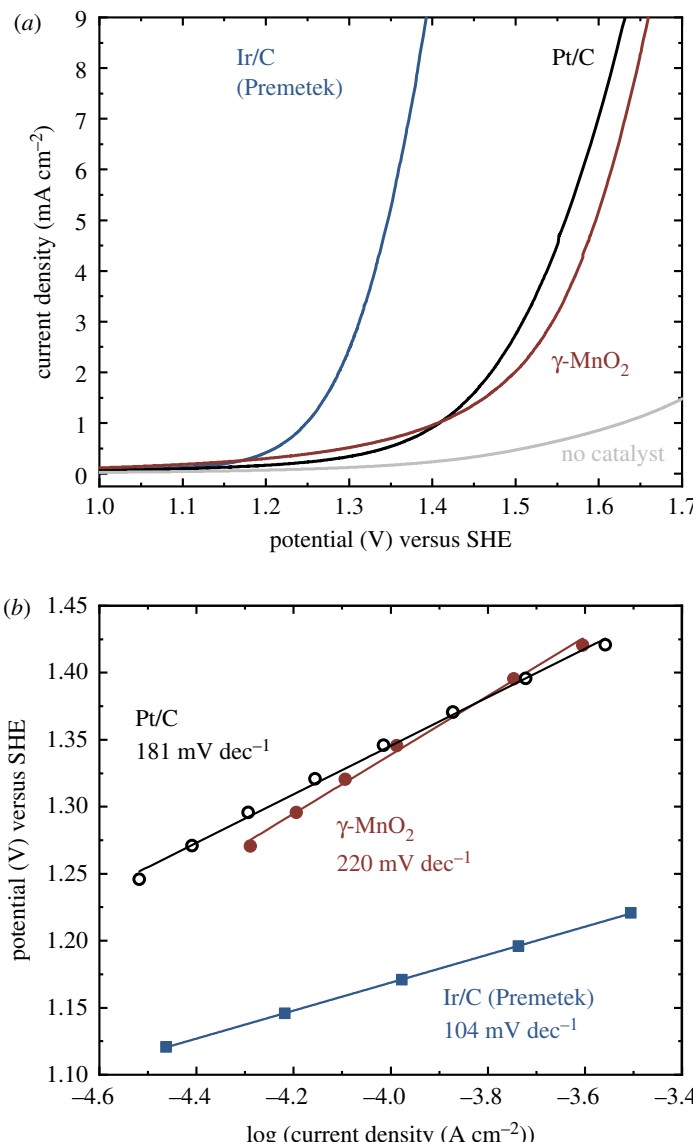

**Figure 6.** (*a*) LSV curves (scan rate: 5 mV s$^{-1}$) and (*b*) Tafel plots of electrodes loaded with the γ-MnO$_2$, Ir/C (Premetek) or Pt/C in the presence of Nafion ionomer. (SHE, standard hydrogen electrode.)

**Table 3.** Comparison of Tafel slopes of the electrodes in the presence and absence of Nafion ionomer in neutral aqueous electrolyte.

| | Tafel slope (mV dec$^{-1}$) | |
|---|---|---|
| catalyst | with Nafion | without Nafion |
| MnO$_2$ | 220 | ∼120 [36,58] |
| IrO$_x$ | 104 | 30 − 60 [6,9] |
| Pt | 181 | 145 (bulk), 210 (nanoparticles) [6] |

The OER activity of the Mn oxides was also evaluated in acidic aqueous electrolyte in the absence of Nafion ionomer, in a similar way to previous studies to find a suitable earth-abundant catalyst for PEM electrolysers [37,38,40,41,45]. In this case, crystal structure dependence of the OER activity was evident (figure 7), which is in contrast with the results of the PEM electrolysers (figure 5). Specifically, the activity of the α-MnO$_2$ was the highest, followed by the β-MnO$_2$, δ-MnO$_2$ and γ-MnO$_2$. The higher activity of

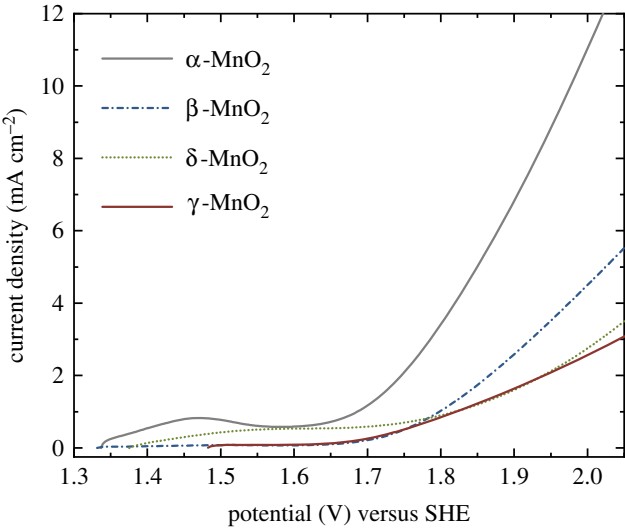

**Figure 7.** LSV curves of the Mn oxide samples synthesized by industrial processes in 0.5 M Na$_2$SO$_4$ aqueous solution (pH 0.2 adjusted by addition of H$_2$SO$_4$) (scan rate: 10 mV s$^{-1}$). The amount of deposited Mn oxides was approximately 0.14 mg cm$^{-2}$.

α-MnO$_2$ than β-MnO$_2$ and δ-MnO$_2$ for the electrochemical OER in acidic aqueous electrolyte is consistent with the report by Stahl and co-workers [62] though Nafion was mixed with the catalysts in their case. According to the report by Stahl and co-workers [62], the comparison of the OER activity of the Mn oxides is largely affected by the oxidation method to drive the OER (i.e. by chemical oxidants, photo-sensitizers or electrodes) and the environment of the catalysts such as pH. Nonetheless, the difference between the OER activity of Mn oxides in PEM electrolysers (figure 5) and in an aqueous electrolyte (figure 7 and table 2) highlights the importance of the evaluation of OER catalysts in PEM electrolysers when one aims to find a suitable catalyst for this type of energy conversion device. Moreover, it was found that the presence of Nafion ionomer also affected the Tafel slope (figure 6b). Here, the Mn oxides showed similar activity in PEM electrolysers (figure 5) even though we observed a clear crystal structure dependence in an aqueous acidic electrolyte (figure 7). This result may indicate that the surface chemical structure of the Mn oxide samples became similar when they were mixed with Nafion ionomer.

## 3.4. Stability of PEM electrolyser

Figure 8 shows the time-dependence of voltage during the electrolysis using the PEM electrolysers at the current density of 0.5 mA cm$^{-2}$. The voltage increased to 1.91 V after 90 min when the γ-MnO$_2$ was used, while the voltage was stable at approximately 1.48 V in the case of Ir/C during initial 90 min. The voltage also increased even in the case of noble metal Pt/C. After 15 h of electrolysis, the detachment of the anode from the MEA was also observed both for the Mn oxide and Pt/C (figure 8). In the time course of more than 24 h, the gradual voltage increase was observed even when Ir/C was used (figure 8).

The similar time course with the rapid voltage increase for the PEM electrolysers with the Mn oxide and Pt/C is an indication that the deactivation processes other than the corrosion of the OER catalysts themselves may cause the increase in the voltage for electrolysis. Here, judging from the anodic current observed from approximately 1.7 V in the case of the PEM electrolyser without OER catalysts (i.e. with only the carbon black) (figure 5), the oxidation of the carbon materials and resulting detachment of catalysts [63,64] could have proceeded together with the electrochemical water splitting reactions in the PEM electrolysers with the Mn oxide and Pt/C, where the initial voltage was slightly larger than 1.7 V (figure 8).

Therefore, the decrease in the required potential for water oxidation is one of the important challenges not only to improve the voltage efficiency for water electrolysis, but also to solve the low stability problem of the PEM electrolysers. To this end, it is worth noting the recent *in situ* spectroscopic study that addressed that the lager Tafel slope of Mn oxides than Ir oxides OER catalysts is attributable to the difference in the rate-determining step (RDS) [61]. Based on the *in situ* spectroscopic identification of the intermediate species for the OER catalysed by Mn and Ir oxides, it was demonstrated that the generation of Mn$^{3+}$ by the oxidation of Mn$^{2+}$ serves as the RDS for the OER by Mn oxides [61],

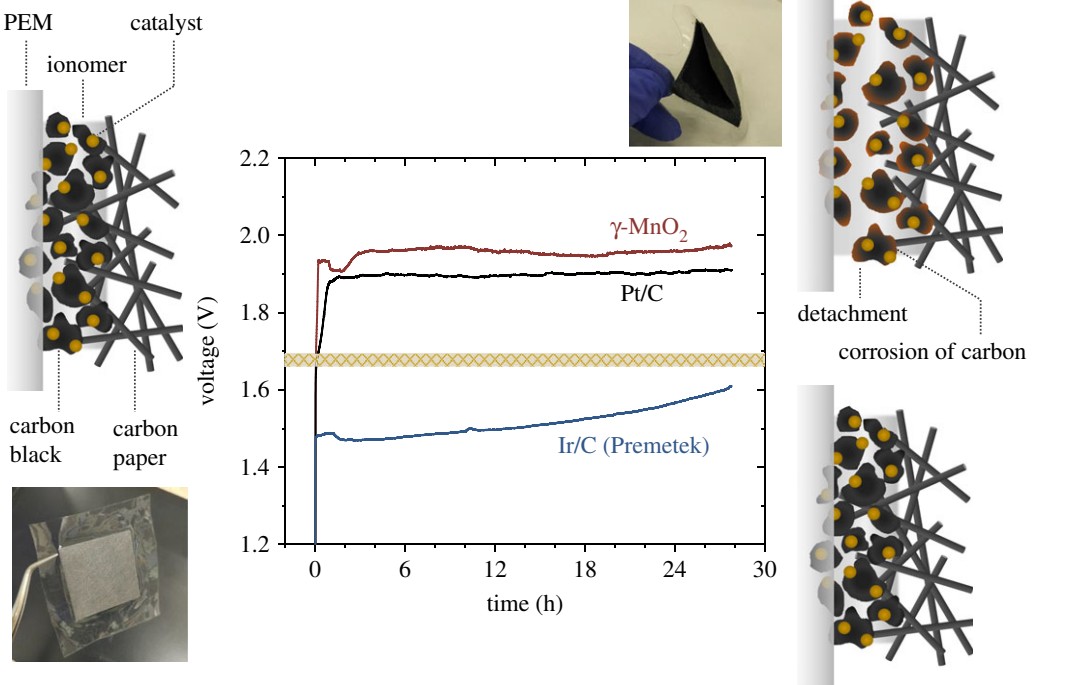

**Figure 8.** Time-dependence of voltage during electrolysis at the current density of 0.5 mA cm$^{-2}$ using the PEM electrolysers with the γ-MnO$_2$, Ir/C (Premetek) or Pt/C as the OER catalyst and schematic of the structure of MEA and possible deactivation processes. The cell resistance (3.42 Ω with γ-MnO$_2$, 1.68 Ω with Pt/C and 3.81 Ω with Ir/C, measured before the electrolysis) was compensated. Corrosion of carbon and detachment of the catalyst may be severe when the voltage is more than approximately 1.7 V (above the brown region in the graph). Photos of the MEA with the γ-MnO$_2$ before and after the electrolysis for 15 h are also shown.

which is in sharp contrast to facile charge accumulation before a chemical RDS for Ir oxides [9,10]. When the concentration of the precursor of the RDS is potential dependent, the Tafel slope becomes smaller than the case without such potential dependence [9]. On the other hand, the expected Tafel slope is 120 mV dec$^{-1}$ when there are no preceding electrochemical steps involving the precursor of the RDS, in agreement with the previously reported Tafel slopes of MnO$_2$ [10,36,59]. Thus, if it is possible to make the RDS of the OER by Mn oxides a chemical step following charge accumulation, by stabilizing Mn$^{3+}$ and facilitating charge accumulation, the Tafel slope is expected to decrease. Recently, several methods to stabilize Mn$^{3+}$ during the OER by Mn oxides have been reported, such as the introduction of nitrogen ligands to make the asymmetry of the ligand field [14], induction of concerted proton-electron transfer [16] and nano-structuring [65]. Application of such strategy to the Mn oxides in PEM electrolysers will decrease not only the overpotential for the OER, but also the Tafel slope, contributing to the resolution of stability problems.

In addition to the optimization of the activity of the catalytic sites, it is noted again that carbon supports have been used for short-term tests or the characterization of electrocatalysts [20], not for long-term electrolysis. If temperature is higher than 50°C, corrosion effects of carbon materials are remarkable, which inhibits the high-temperature PEM operation required to increase the proton conductivity of Nafion membrane [48]. Thus, substitution of carbon supports with oxide one will be the next step.

In the present study, the Mn oxide samples were simply mechanically mixed with the carbon black, and the SEM images showed the non-homogeneous aggregation of the Mn oxide and carbon black nanoparticles (electronic supplementary material, figure S1). As various factors, such as the cell resistance and the catalysts loading, remain to be optimized, it is expected that there is significant room for improvement of the overall performance of PEM electrolysers, and such attempts are currently underway in our laboratory.

## 4. Conclusion

We have compared the OER activity of Mn oxides, Ir and Pt in the same condition in PEM electrolysers. As a result, we have demonstrated that the Mn oxide samples synthesized by industrial processes can

exhibit comparable activity for the OER to that of Pt/C in PEM electrolysers. Also, the activity trends of the Mn oxides evaluated in an acidic aqueous electrolyte and in PEM electrolysers were different, demonstrating the importance of the evaluation of OER catalysts in a real device condition of PEM electrolysers. The increase in voltage was observed during electrolysis at a constant current density in the case of PEM electrolysers not only when the Mn oxides were used but also when Pt/C was used. The observed deactivation is distinct from the stable long-term operation with Ir/C and is, because it was observed even when Pt/C was used, probably attributed to the oxidation of carbon supports due to the competing oxidation of carbon material under high-voltage conditions. To solve the stability issues, not only decreasing the overpotential for the OER by Mn oxides, but lowering the larger Tafel slopes based on the regulation of charge accumulation processes could be of potential importance for future development of the Mn oxide OER catalysts.

Data accessibility. There are no additional data to accompany this manuscript and the electronic supplementary material. All relevant datasets are within the main body of the manuscript or the electronic supplementary material.

Authors' contributions. T.H., N.B.-M., A.Y. and K.S. performed experiments. T.H., N.B.-M., A.Y., K.S. and R.N. analysed data. T.H. and R.N. wrote the manuscript.

Competing interests. We declare we have no competing interests.

Funding. This work was supported by JSPS Grant-in-Aid for Scientific Research no. 26288092. T.H. was also supported by Grant-in-Aid for JSPS Fellows no. 15J10583.

Acknowledgements. We are grateful to two anonymous reviewers.

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
