## [Reviewer comments · Royal Society Open Science]

Review History

RSOS-190122.R0 (Original submission)

Review form: Reviewer 1

Is the manuscript scientifically sound in its present form?

Yes

Are the interpretations and conclusions justified by the results?

Yes

Is the language acceptable?

Yes

Is it clear how to access all supporting data?

Yes

Do you have any ethical concerns with this paper?

No

Have you any concerns about statistical analyses in this paper?

No

Recommendation?

Accept with minor revision (please list in comments)

Comments to the Author(s)

Nakamura et al have reported a series of Manganese polymers that act as cost-effective catalysts for Oxygen Evolution Reaction (OER). The manuscript may be accepted with the answers to the following queries.

1. In PEM electrolysis, the gamma-MnO₂ showed higher current density compared to other Mn oxides. The authors should give a plausible reason for this.
2. When the OER activity of the Mn oxides is evaluated in the acidic environment in the absence of Nafion ionomer, the alpha-MnO₂ showed highest activity compared to other oxides. What is the reason for this behaviour?

Review form: Reviewer 2

Is the manuscript scientifically sound in its present form?

Yes

Are the interpretations and conclusions justified by the results?

Yes

Is the language acceptable?

Yes

Is it clear how to access all supporting data?

Yes

Do you have any ethical concerns with this paper?

No

Have you any concerns about statistical analyses in this paper?

Yes

Recommendation?

Accept with minor revision (please list in comments)

Comments to the Author(s)

This is a timely article about water splitting, which in my opinion should be accepted. I only find a number of small language problems on p2: Line 1. Water splitting Line 6: current standard PEMs... Line 22: substitutes for...

In figs 4-8 the greek letters alpha etc are missing in my manuscript.

Decision letter (RSOS-190122.R0)

29-Mar-2019

Dear Dr Nakamura:

Title: Electrochemical Characterization of Manganese Oxides as a Water Oxidation Catalyst in PEM Electrolyzers
Manuscript ID: RSOS-190122

Thank you for submitting the above manuscript to Royal Society Open Science. On behalf of the Editors and the Royal Society of Chemistry, I am pleased to inform you that your manuscript will be accepted for publication in Royal Society Open Science subject to minor revision in accordance with the referee suggestions. Please find the reviewers' comments at the end of this email.

The reviewers and handling editors have recommended publication, but also suggest some minor revisions to your manuscript. Therefore, I invite you to respond to the comments and revise your manuscript.

Please also include the following statements alongside the other end statements. As we cannot publish your manuscript without these end statements included, if you feel that a given heading is not relevant to your paper, please nevertheless include the heading and explicitly state that it is not relevant to your work. We have included a screenshot example of the end statements for reference.

- Ethics statement

Please clarify whether you received ethical approval from a local ethics committee to carry out your study. If so please include details of this, including the name of the committee that gave consent in a Research Ethics section after your main text. Please also clarify whether you received informed consent for the participants to participate in the study and state this in your Research Ethics section.

OR

Please clarify whether you obtained the necessary licences and approvals from your institutional animal ethics committee before conducting your research. Please provide details of these licences and approvals in an Animal Ethics section after your main text.

OR

Please clarify whether you obtained the appropriate permissions and licences to conduct the fieldwork detailed in your study. Please provide details of these in your methods section.

- Acknowledgements

Because the schedule for publication is very tight, it is a condition of publication that you submit the revised version of your manuscript before 07-Apr-2019. Please note that the revision deadline will expire at 00.00am on this date. If you do not think you will be able to meet this date please let me know immediately.

Best wishes,
Dr Laura Smith
Publishing Editor, Journals

On behalf of the Subject Editor Professor Anthony Stace and the Associate Editor Professor Claire Carmalt.

RSC Associate Editor:
Comments to the Author:

(There are no comments.)

RSC Subject Editor:

Comments to the Author:

(There are no comments.)

Reviewer comments to Author:

Reviewer: 1

Comments to the Author(s)

Nakamura et al have reported a series of Manganese polymers that act as cost-effective catalysts for Oxygen Evolution Reaction (OER). The manuscript may be accepted with the answers to the following queries.

1. In PEM electrolysis, the gamma-MnO₂ showed higher current density compared to other Mn oxides. The authors should give a plausible reason for this.

2. When the OER activity of the Mn oxides is evaluated in the acidic environment in the absence of Nafion ionomer, the alpha-MnO₂ showed highest activity compared to other oxides. What is the reason for this behaviour?

Reviewer: 2

Comments to the Author(s)

This is a timely article about water splitting, which in my opinion should be accepted. I only find a number of small language problems on p2: Line 1. Water splitting Line 6: current standard PEMs... Line 22: substitutes for...

In figs 4-8 the Greek letters alpha etc are missing in my manuscript.

Author's Response to Decision Letter for (RSOS-190122.R0)

See Appendix A.

Decision letter (RSOS-190122.R1)

24-Apr-2019

Dear Dr Nakamura:

Title: Electrochemical Characterization of Manganese Oxides as a Water Oxidation Catalyst in PEM Electrolyzers

Manuscript ID: RSOS-190122.R1

It is a pleasure to accept your manuscript in its current form for publication in Royal Society

Open Science. The chemistry content of Royal Society Open Science is published in collaboration with the Royal Society of Chemistry.

RSC Associate Editor
Comments to the Author:
The revisions are sufficient. The manuscript can be accepted.

Reviewer(s)' Comments to Author:

Appendix A

Response to Referees

Title: Electrochemical Characterization of Manganese Oxides as a Water Oxidation Catalyst in PEM Electrolyzers

Manuscript ID: RSOS-190122

As to the comments of Reviewer #1:

Comment 1-1.

In PEM electrolysis, the gamma-MnO₂ showed higher current density compared to other Mn oxides. The authors should give a plausible reason for this.

Reply 1-1.

We would like first to express our appreciation to your comments and questions.

In this study, as mentioned at the end of Section 4.3, we understand that we found the Mn oxides showed similar activity in PEM electrolyzers (Figure 5), even though we observed a clear crystal structure dependence in an aqueous acidic electrolyte (Figure 7). We expect that this result may indicate that the surface chemical structure of the Mn oxide samples became similar when they were mixed with Nafion ionomer.

Nonetheless, we agree that it is important to clarify the reason why we observed the highest current density when we used γ -MnO₂. By the clarification, we can find a way to optimize the catalyst.

Here, we note the stable potential window of γ -MnO₂ in an acidic condition (A. Li *et al.*, *Angew. Chem. Int. Ed.* **2019**, 58, 5054–5058). Although many parameters affect the performance of PEM electrolysis, we expect that the existence of such a stable potential window in acid may contribute the high OER activity of γ -MnO₂. Thus, we have added sentences as follows.

Sentences added (From the 2nd line from the bottom, Page 4):

Investigation of the reason of the highest activity and the optimization of the material are currently underway. However, it is to be noted that γ -MnO₂ has a stable potential window in an acidic condition, where the OER can be catalyzed efficiently by suppressing the corrosion reaction (A. Li *et al.*, *Angew. Chem. Int. Ed.* **2019**, 58, 5054–5058). Therefore, we expect that the existence of such a stable potential window in acid may contribute the high OER activity of γ -MnO₂.

Comment 1-2.

When the OER activity of the Mn oxides is evaluated in the acidic environment in the absence of Nafion ionomer, the alpha-MnO₂ showed highest activity compared to other oxides. What is the reason for this behaviour?

Reply 1-2.

At the present stage, we could not answer the comment. To obtain the rational reasons for the different activity trend observed in the systems with and without Nafion ionomer, spectroscopic characterization at the interface of MnO₂ and Nafion is needed. Such studies are currently under investigation and will be reported elsewhere. Therefore, no revision was made on this comment.

As to the comments of Reviewer #2:

Comment 2-1.

I only find a number of small language problems on p2: Line 1. Water splitting Line 6: current standard PEMs... Line 22: substitutes for...

In figs 4-8 the greek letters alfa etc are missing in my manuscript.

Reply 2-1.

We sincerely appreciate the comments from the reviewer. We have corrected accordingly.